# Combined Use of a Transmission Detector and an EPID-Based In Vivo Dose Monitoring System in External Beam Whole Breast Irradiation: A Study with an Anthropomorphic Female Phantom

**Chiara Arilli [1], Yannik Wandael [2], Chiara Galeotti [2], Livia Marrazzo [1], Silvia Calusi [3], Mattia Grusio [4], Isacco Desideri [3], Franco Fusi [3], Angelo Piermattei [4], Stefania Pallotta [1,3] and Cinzia Talamonti [1,3,*]**

1   Medical Physics Unit, AOU Careggi, 50134 Florence, Italy; chiara.arilli@unifi.it (C.A.); livia.marrazzo@unifi.it (L.M.)
2   Radiotherapy Unit, AOU Careggi, 50134 Florence, Italy; yannickwandael@gmail.com (Y.W.); galeottic@aou-careggi.toscana.it (C.G.)
3   Department of Experimental and Clinical Biomedical Sciences "Mario Serio", University of Florence, 50134 Florence, Italy; silvia.calusi@unifi.it (S.C.); isacco.desideri@unifi.it (I.D.); franco.fusi@unifi.it (F.F.); stefania.pallotta@unifi.it (S.P.)
4   UOC Fisica Sanitaria, Fondazione Policlinico Universitario Agostino Gemelli, 00168 Rome, Italy; mattiagrusio@gmail.com (M.G.); angelo.piermattei@gmail.com (A.P.)
*   Correspondence: Cinzia.Talamonti@unifi.it

**Abstract:** We evaluate the combined usage of two systems, the Integral Quality Monitor (IQM) transmission detector and SoftDiso software, for in vivo dose monitoring by simultaneous detection of delivery and patient setup errors in whole breast irradiation. An Alderson RANDO phantom was adapted with silicon breast prostheses to mimic the female anatomy. Plans with simulated delivery errors were created from a reference left breast plan, and patient setup errors were simulated by moving the phantom. Deviations from reference values recorded by both monitoring systems were measured for all plans and phantom positions. A 2D global gamma analysis was performed in SoftDiso for all phantom displacements. Both IQM signals and SoftDiso R-values are sensitive to small MU variations. However, only IQM is sensitive to jaw position variations. Conversely, IQM is unable to detect patient positioning errors, and the R-value has good sensitivity to phantom displacements. A gamma comparison analysis allows one to determine alert thresholds to detect phantom shifts or relatively large rotations. The combined use of the IQM and SoftDiso allows for fast identification of both delivery and setup errors and substantially reduces the impact of error identification and correction on the treatment workflow.

**Keywords:** quality assurance; in vivo dosimetry; transmission radiation detectors; whole breast irradiation

## 1. Introduction

After breast conserving surgery, radiotherapy is the usual therapeutic strategy for breast cancer patients providing, with fewer toxicity effects, equivalent survival outcomes when compared with mastectomy [1,2]. Conventional three-dimensional conformal radiotherapy (3DCRT), with tangential external photon beams, is the standard irradiation modality, which is known to successfully ensure the local control of the disease [2,3]. Moreover, in recent years, advanced techniques like intensity modulated radiation therapy (IMRT), helical tomotherapy (HT) and volumetric modulated arc

therapy (VMAT) were investigated and compared in terms of normal tissue toxicities and dosimetric advantages [4–8]. However, the preferred breast irradiation modality differs between radiotherapy departments. At our institution, breast cancer is the most recurrent pathology treated with radiotherapy, and its high incidence encouraged us to build and share a method to improve the dose delivery accuracy in 3DCRT breast treatments. Often in radiotherapy, patient quality assurance programs (QA) are performed in pre-treatment modality. However, pre-treatment verification presents several drawbacks: (1) the actual patient geometry is never included in pre-treatment measurements, making it difficult to estimate the effect of observed dosimetric deviations on the actual patient; (2) deviations from intended treatment occurring during the dose delivery cannot be detected [9,10]; (3) pre-treatment verification requires additional measurement sessions, taking up valuable linac time and increasing workload. The first issue is usually addressed in commercial systems by reworking observed deviations on the patient anatomy (using the planning CT scan) [11–13]; however, the last two issues cannot be circumvented by using pre-treatment verification. To solve these questions, in vivo dosimetry systems (IVDs) that perform dose measurements during the treatment and compare them to the intended dose are sometimes introduced into the clinical routine. In this work, a study of uncertainties affecting breast irradiation was carried out. Three main error sources were analyzed: delivery errors, setup errors and human errors. Delivery errors are systematic errors affecting delivery parameters which can lead to output variations (e.g., incorrect linac dose calibration or incorrect calibration of the multi-leaf collimator or of the jaws). Inter-fraction setup errors instead are more difficult-to-control errors tightly related to the patient alignment procedure followed by operators, to anatomical modifications and to treatment preparation imaging devices. Regrettably, human mistakes could also occur, e.g., incorrect arrangement of the patient, perhaps due to clothes or objects accidentally located inside the treatment field, or incorrect identification of the patient occurs. The effectiveness of IVDs in detecting systematic and random treatment errors has been investigated in several review papers and reports [9,14–19]. In particular, all authors agreed that standard quality pre-treatment checks are not able to detect a large number of treatment errors, highlighting the need for IVDs. Recently, new international recommendations, following the suggestions indicated in many incident reports [20–23], have encouraged the introduction of IVDs as good practice in radiotherapy [24–29]. For example, article 63 of the European Directive 2013/59/Euratom [24] clearly states that the member countries have to include a risk analysis in their quality management systems for radiotherapy practice. In some European countries, such as Sweden, Denmark, Norway, the Czech Republic, and France, an IVD is mandatory [27–29]. Commercial systems for in vivo dose monitoring are available with different objectives. Several devices measure the patient's dose by using detectors placed on the patient (for example, on the patient's skin) or next to the patient (in a known position), or by processing the radiation transmitted through the patient. Other commercial systems measure the photon fluence at the linac output. Real-time in vivo devices, such as diodes, metal-oxide semiconductor field effect transistors (MOSFETs) and electronic portal imaging devices (EPID) [30,31], belong to the first group. In particular, EPID systems use back projection methods on measured portal images to obtain 2D [32–34] or 3D [35,36] patient dose distributions directly comparable with the planned dose. Transmission radiation detectors (TRDs), consisting usually of an array located on the linac head, belong instead to the second group. However, these devices monitor the delivery but not the patient dose; therefore, they cannot be classified as IVDs [37–42]. Our project aims to implement a new quality assurance (QA) procedure for patients undergoing highly conformal radiotherapy treatments that combines a transmission detector to monitor the fluence entering the patient and in vivo dosimetry with an EPID to measure the dose actually released into the patient during the treatment. An Integral Quality Monitor detector (IQM, iRT Systems GmbH, Koblenz, Germany) and a SoftDiso (Best Medical Srl, Italy) system have been simultaneously operated to evaluate their sensitivity in regard to detecting small delivery and setup errors for 3DCRT breast irradiation of a female phantom.

## 2. Materials and Methods

### 2.1. Equipment

In this study, 6MV energy photon beams were delivered with Precise and Synergy BM linacs (Elekta AB, Stockholm, Sweden), both equipped with an iViewGT a-Si panel EPID (Elekta AB, Stockholm, Sweden). Precise and Synergy multi-leaf collimators provide maximum field sizes of $40 \times 40$ cm$^2$ and $16 \times 21$ cm$^2$ respectively, both using two banks of 40 leaves. The Synergy linac is also equipped with an advanced robotic patient positioning platform (HexaPOD evo RT system) enabling sub-millimeter couch movements with six degrees of freedom. A scheme of the experimental setup is shown in Figure 1.

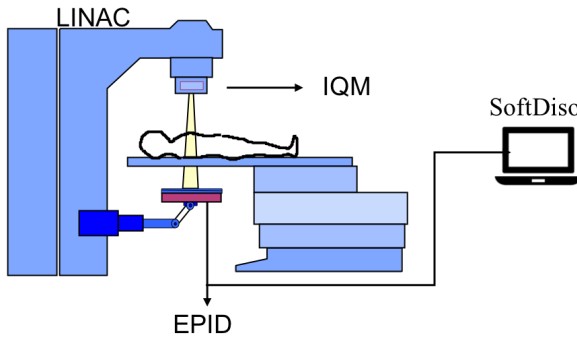

**Figure 1.** A scheme of measurement setup is displayed.

### 2.2. Monitoring Devices

The IQM device is an online delivery monitoring system, checking in real time the accuracy of the radiation fluence. The device consists of a large area transmission chamber, designed to be mounted on the linac head. An electric field gradient is generated by the inclined chamber electrode plates, providing a non-uniform charge collection linearly varying in the direction parallel to the multi-leaf collimator's motion. During the irradiation, a spatially sensitive dose-area product is generated and reported in arbitrary units (counts). A picture of an IQM detector mounted on a linac head is shown in Figure 2.

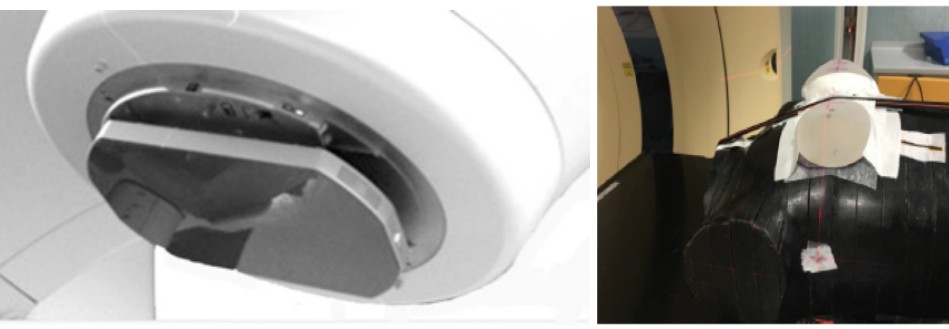

**Figure 2. Left**. The Integral Quality Monitor detector (IQM) system mounted on the linac head. **Right**. Alderson RANDO phantom with two silicon prostheses to simulate the female anatomy.

The IQM software allows an easy comparison of the acquired measurements with reference data to detect possible deviations from nominal parameters. The system has also an integrated inclinometer to measure collimator and gantry angles. Details of the detector described by several authors can be found in reference [40].

The SoftDiso system is a piece of online monitoring software reconstructing the dose by using the transmission signal measured by an EPID device. The reconstructed dose is compared with the

planned dose through an R-value, defined as the ratio of the measured to calculated doses at the isocenter point (Diso/DTPS) [32,33]. To check the inter-session patient position reproducibility and to identify possible anatomical variations over the whole treatment, a 2D gamma-index analysis tool compares an EPID reference image (usually acquired during the first/second session) with treatment images by using global gamma passing rate ($\gamma_{PR}$) and gamma mean value ($\gamma_{mean}$) indexes.

### 2.3. Phantom

A suitable phantom was made to reproduce the real female anatomy. An Alderson RANDO phantom was modified with two silicon breast prostheses placed on the phantom chest. Ultrasound gel was used to reduce the air gap at the prosthesis–chest interface.

A CT scan of the phantom was acquired with a Big Bore Philips scanner (Philips Medical Systems, Fitchburg, WI). Two reference 3DCRT plans of the phantom left breast were created for 50 Gy dose prescription (25 fractions) with Philips Pinnacle3 Professional TPS version 9.10 (Philips Medical Systems, Fitchburg, WI) for Precise and Synergy linacs respectively. Figure 3 shows the dose distribution for the Precise plan. The clinical target volume (CTV), the heart, the ipsilateral and controlateral lungs, the contralateral breast and the spinal cord were defined on the CT phantom.

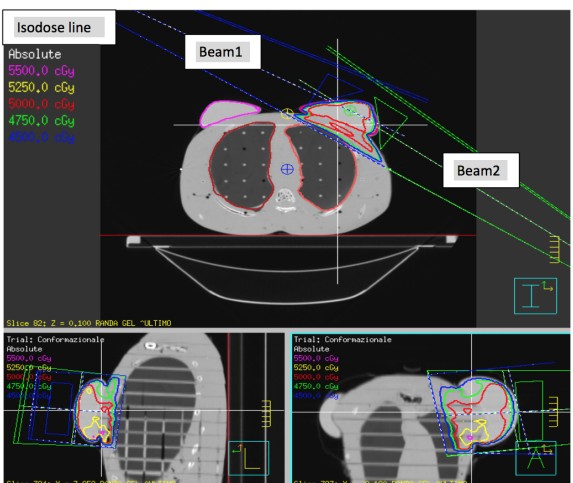

**Figure 3.** A screenshot of reference treatment plan dose distribution for 50 Gy dose prescription. Isodose lines are displayed in sagittal, coronal and axial views.

Plan parameters are reported in Table 1.

**Table 1.** Parameters of the two simulated treatment plans used in the study.

| Precise | Beam 1 | Beam 2 |
|---|---|---|
| Gantry angle | 120° | 295° |
| Collimator angle | 264° | 96° |
| Field size (cm$^2$) | 14 × 11 | 14 × 11 |
| Beam energy (MV) | 6 | 6 |
| MU | 177 | 191 |
| Wedge | yes | yes |
| **Synergy** | **Beam 1** | **Beam 2** |
| Gantry angle | 300° | 125° |
| Collimator angle | 96° | 264° |
| Equivalent field size (cm$^2$) | 14 × 11 | 14 × 11 |
| Beam energy (MV) | 6 | 6 |
| MU | 161 | 143 |
| Wedge | yes | yes |

### 2.4. Short-Term Reproducibility of Systems

The short-term reproducibility of SoftDiso R-value was checked by repeating the reference Precise plan 20 times and was evaluated by the ratio of the standard deviation to the average value ($\sigma_R / R_{avg}$). The short-term reproducibility of IQM signal has already been studied at our department [43].

### 2.5. Simulated Delivery Errors

To evaluate the sensitivity of the two devices to delivery errors, 12 plans for Precise linac were created by modifying the reference plan parameters. To simulate output errors, derived, for example, for an incorrect dose calibration of linac, MUs were modified by adding 2 MU, 3 MU, 5 MU and 10 MU at nominal collimator positions; to mimic an incorrect calibration of jaws, the jaw located near to the ipsilateral lung was displaced by $\pm2$ mm, $\pm3$ mm, $\pm5$ mm and $\pm7$ mm, at nominal MU. IQM and SofDiso devices were operated simultaneously during the delivery of all 13 plans (the reference and the 12 modified). The average deviations of the IQM signals and R-values from the reference plan values were evaluated in 5 consecutive measurement sessions.

### 2.6. Simulated Setup Errors

Setup errors were simulated with Synergy couch by displacing the phantom by 2 mm, 3 mm, 5 mm, 7 mm and 10 mm in anterior, lateral and longitudinal directions. The phantom was also rotated around the longitudinal axis by $1°$ and $2.8°$ (maximum extent of rotation reachable with HexaPOD system). The average deviations of R-value from reference values were evaluated in five consecutive measurement sessions. The $\gamma_{\mathrm{PR}}$ and the $\gamma_{\mathrm{mean}}$ values were calculated with SoftDiso for all images and compared with the corresponding values obtained with the phantom in the reference position. For all cases, $\gamma_{\mathrm{PR}}$ and $\gamma_{\mathrm{mean}}$ were evaluated with 2%/2 mm global criteria. Alert thresholds to detect shifts of 2 mm were established by using the gamma comparison. With the same thresholds we also studied the sensitivity to rotations.

## 3. Results

### 3.1. Short-Term Reproducibility of Systems

The R-value short-term reproducibility ($\sigma_{\mathrm{R}/\mathrm{R}_{avg}}$) was found to be 0.6%, while the IQM signal was measured to be reproducible within 0.08%, consistent with our previous study [43].

### 3.2. Simulated Delivery Errors

The average deviations from reference counts and R-values of simultaneously operated IQM and SoftDiso due to simulated delivery errors obtained from the five measurement runs are reported in Figure 4. Both SoftDiso and IQM detected all MU variations showing excellent linearity and sensitivities of $(0.53 \pm 0.13)$%/MU and $(0.53 \pm 0.01)$%/MU respectively. A remarkable correlation between the outputs of the two devices was also observed. Furthermore, the IQM is also able to detect small jaw position variations, showing again, good sensitivities of $(1.17 \pm 0.02)$%/mm and $(1.05 \pm 0.02)$%/mm in both closing and opening movement directions respectively. The linearity in this case is only approximate, pointing either to an intrinsically non-linear response or to an underestimation of measurement errors. Nevertheless, IQM measurement errors are negligible compared to deviations of signal consequent to delivery errors in clinical routine. On the contrary, the R-value was found to be almost insensitive to the same relatively small jaw position variations, while still showing a reasonable response linearity, with large uncertainty though.

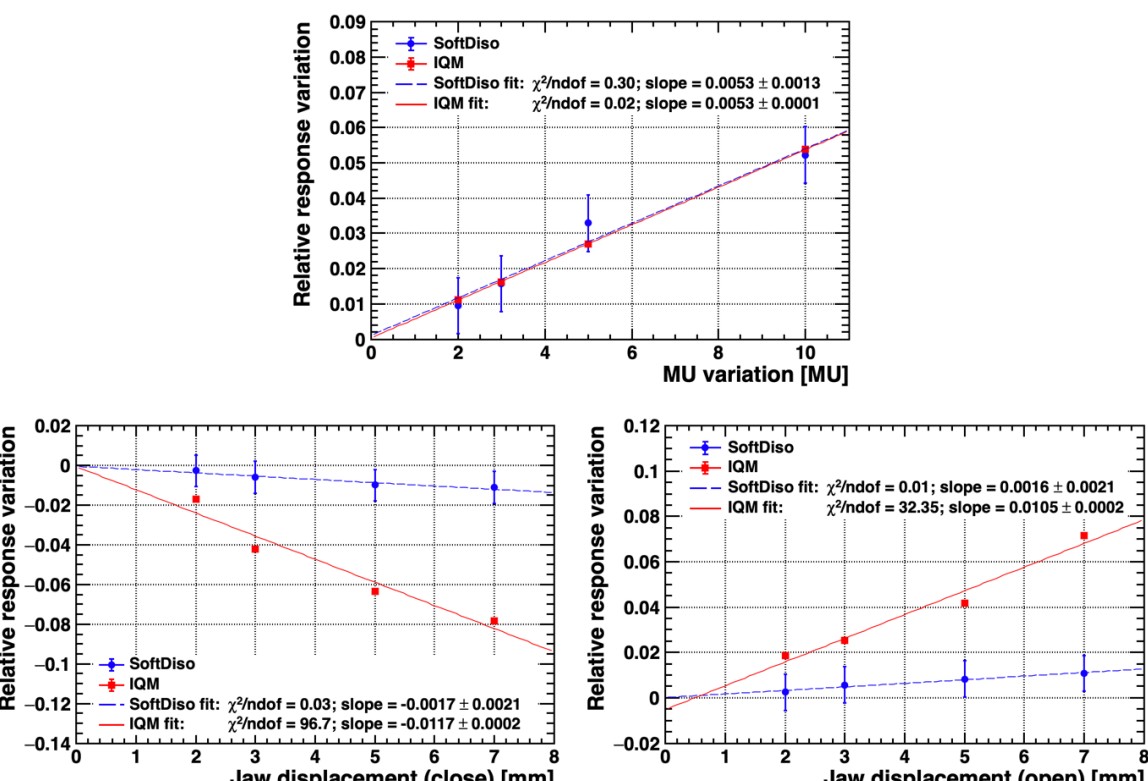

**Figure 4.** Deviations from reference values of IQM signal and SoftDiso R-values as functions of MU variations (**top**) and of jaw position variation in close (**bottom left**) and open (**bottom right**) directions.

### 3.3. Simulated Setup Errors

The IQM system is intrinsically insensitive to patient positioning, being placed upstream of the patient along the beam line. The SoftDiso provides, therefore, the only tool available to check for the correct patient positioning. The average R-value deviations (absolute values) for all phantom displacements and rotations with respect to the reference position are shown in Figure 5. Given the characteristics of the beams and the irradiation region considered in this study, the R-value measurements show good linearity and a sensitivity of $(0.74 \pm 0.12\%)/\text{mm}$ to displacements in the anterior direction, where the strictest control of patient positioning is required. In contrast, as expected, the R-value is insensitive to movements in lateral and longitudinal directions, and to rotations, which are less critical in this specific case.

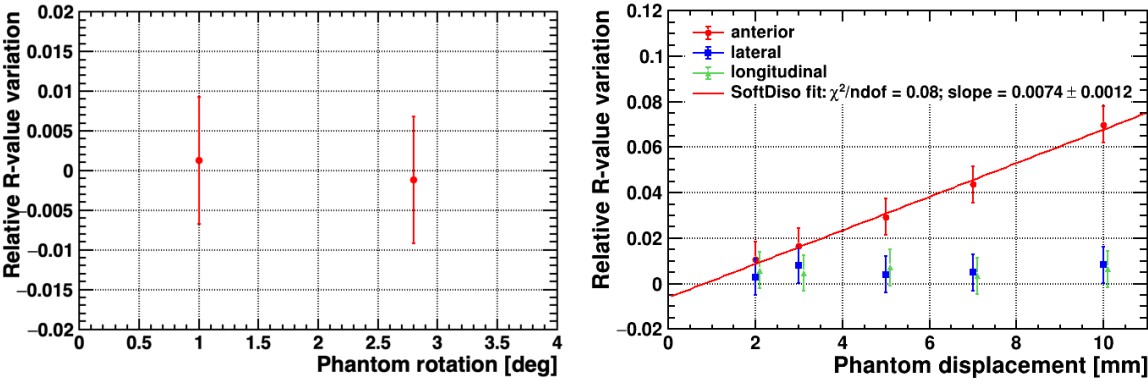

**Figure 5.** Deviations from reference values of the SoftDiso R-value for phantom rotations (**left**) and translations (**right**).

To further explore the sensitivity of SoftDiso to patient positioning errors, a gamma-index comparison of EPID images with displaced phantom was performed. The results for 2%/2mm global criteria are presented in Figure 6. To reveal phantom shifts $\geq$ 2 mm, alert thresholds of $\gamma_{PR} < 89.7\% \pm 0.4\%$ and $\gamma_{mean} > 0.32 \pm 0.04$ were obtained. The $\gamma_{PR}$ and $\gamma_{mean}$ indexes show good linearity with phantom displacements. Results of the comparison for phantom rotations are reported in Table 2.

From the measurements we conclude that the sensitivity to small rotations is difficult to asses, with $\gamma_{PR}$ remaining within the alert threshold for both rotation angles and $\gamma_{mean}$ going slightly above-threshold for the maximum simulated rotation of 2.8°.

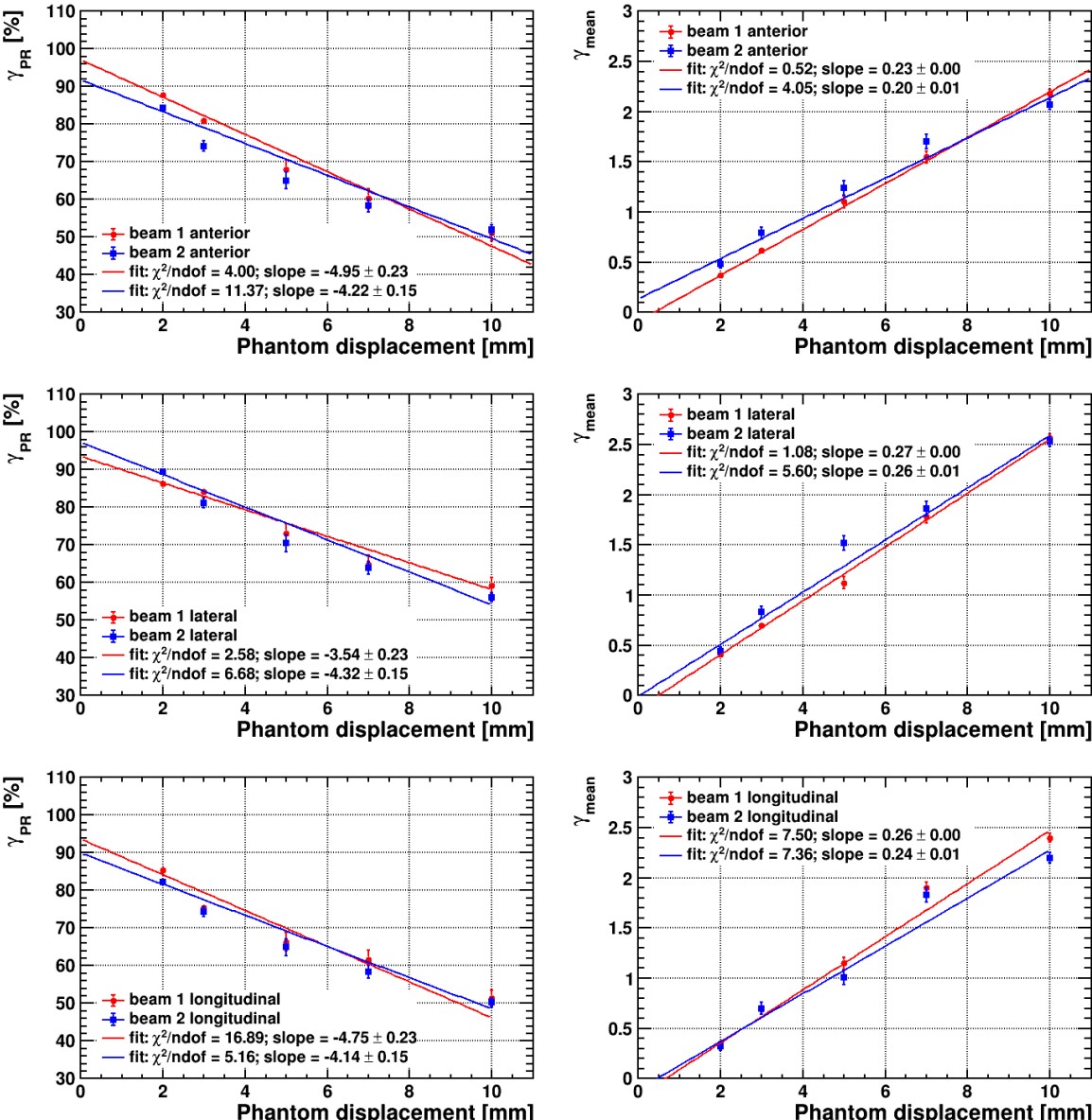

**Figure 6.** Measured $\gamma_{PR}$ and $\gamma_{mean}$ indexes as functions of phantom anterior (**top**); lateral (**middle**) and longitudinal (**bottom**) displacements for both beam 1 and beam 2 used in the simulated treatment plan.

**Table 2.** Results of the gamma-index comparison for phantom rotations.

| Rotation Angle | $\gamma_{PR}$ Beam 1 | $\gamma_{mean}$ Beam 1 | $\gamma_{PR}$ Beam 2 | $\gamma_{mean}$ Beam 2 |
| --- | --- | --- | --- | --- |
| $1.0°$ | $(95.0 \pm 0.4)\%$ | $0.19 \pm 0.05$ | $(95.7 \pm 0.4)\%$ | $0.18 \pm 0.03$ |
| $2.8°$ | $(91.0 \pm 0.6)\%$ | $0.37 \pm 0.05$ | $(91.2 \pm 0.6)\%$ | $0.33 \pm 0.04$ |

## 4. Discussion

In this study deviations in treatment delivery and in patient setup were simulated. Both types of errors happen in clinical routine. Scenarios can be various: small, but even large deviations can occur during the treatment, for example, when a wrong linac dose calibration is done or when patient is not correctly identified or prepared for the therapy. The strength of the combined use of the two systems in detecting all different kinds of errors, even when very small, has been evidenced in this work. Separate use of IQM or SoftDiso devices has been widely discussed for different irradiation modalities [43–49], but to our knowledge, no study exists of their combined use, in particular for breast irradiation. A clinical study of the SoftDiso-based quality assurance procedure for post-mastectomy IMRT and VMAT radiation therapy is reported in reference [50], where specific thresholds for R-value, $\gamma_{PR}$ and $\gamma_{mean}$ indexes were proposed, based on the clinical experience and validated with a phantom chest. The authors confirmed the feasibility and the importance of a QA procedure based on EPID-based IVD to detect inter-fraction setup errors to overcome the weakness of a setup control based only on weekly cone beam CT acquisitions. The high IQM sensitivities to simulated delivery errors have been assessed by several authors [43–46] and confirmed in our work. However, to our knowledge, studies of the application of IQM to breast irradiation have not been reported elsewhere. From our measurements, the IQM device demonstrates high sensitivity to small MU variations and jaw position deviations. The SoftDiso R-value was also found to be able to detect MU variations with a sensitivity slightly smaller than for IQM. However, the R-value is much less sensitive to collimator position variations, as expected given the stability of output factors for small modifications of the field size. The IQM system is obviously insensitive to patient setup errors given its positioning upstream the patient itself. However, in this work we have demonstrated the ability of the EPID-based SoftDiso to detect simulated setup errors. The SoftDiso R-value was shown to have good sensitivity to small displacements in the anterior direction—the error most critical for the particular simulated plan used. To recover full sensitivity to 3D displacements a gamma-index analysis was needed. This analysis, with 2%/2mm criteria, yielded alert thresholds of about 90% for $\gamma_{PR}$ and 0.4 for $\gamma_{mean}$, in agreement with those proposed in the literature [50]. The $\gamma_{mean}$ threshold was also demonstrated to be able to detect a phantom rotation of $2.8°$. It is worth noticing here that the gamma-index analysis, while being fairly sensitive, cannot distinguish between patient positioning errors and beam setup errors. In this study we have therefore shown that the combined and simultaneous use of IQM and an EPID-based system provides a complete monitoring of beam and patient setup errors. Both devices allow one to set alert thresholds, which can be suitably adjusted according to the specific treatment, providing a warning message or sound when the measurement is out of tolerance. The IQM system can generate a real time warning when the delivery is incorrect. On the other hand, the R-values provided by the SoftDiso immediately after the sessions can be saved in a table, showing not only whether the tolerance has been exceeded but also whether a trend over a series of sessions appears, pointing to the need for a setup correction. Finally, the response of gamma analysis comparison is quickly available after a very fast data analysis (few minutes). Consequently, by combining the IQM and the SoftDiso, delivery and/or setup errors are detected quickly during or shortly after the delivery, allowing one to correct the inaccuracies during or immediately following treatment sessions. Both devices can be easily used by the therapist and do not affect the workflow, increase the booking time or involve the physician or physicist except when an alert message is activated. In the latter case, moreover, the combined output of the two devices can help in rapidly identifying the type of error to be corrected. The work presented is a pilot study simulating errors which could occur while delivering a radiotherapy conforming breast

treatment. The clinical implementation and the extension of this quality protocol to other districts is the next step in our project. The combined use of the two detectors is advantageous for IMRT or VMAT treatments too. In these cases, small deviations in positions of the multileaf collimator leaves could toughly affect the output delivery.

## 5. Conclusions

Although the European Commission clearly identified IVD as a relevant element in radiotherapy QA programs, only a few Institutions use it routinely for different reasons. First of all, many of the commercially available systems, although exhibiting good performances, do not respond in real time, often having very long data processing times. Additionally, many devices require dedicated staff, because of their complicated method to enter the data for processing. Finally, often a single system is not able to discriminate between all treatment errors, as demonstrated in this work. In this study we have presented a pilot study showing a new strategy to perform fast and reliable QA tests with a low impact on the workflow of the institution and patient waiting list. The combined use of IQM and SoftDiso devices was demonstrated to be able to efficiently detect small delivery and setup errors in 3DCRT breast irradiation. We have shown that the two devices provide complementary information to detect in almost real time all types of errors. The method would represent an important step forward in the clinical routine to increase the quality of external breast irradiation. Our study concentrated on breast irradiation, but the method could be easily extended to other anatomical districts or treatment modalities, especially when advanced techniques are used. The method proposed helps to comply with the European Commission requirements [24], and represents a new strategy in QA programs, thereby overcoming the drawbacks of conventional pre-treatment verification, which are very time consuming and ineffective to detect in-treatment errors.

**Author Contributions:** Methodology, formal analysis and Writing–original draft preparation: C.A.; Funding acquisition: F.F. and S.P.; Investigation: Y.W. and C.G.; Software: M.G. and A.P.; Validation: L.M.; Visualization: S.C. and I.D.; Conceptualization, Supervision, Writing–review and editing: C.T. All authors have read and agreed to the published version of the manuscript.

**Funding:** This work was supported by Italian MIUR (the Italian Ministry of Education, University and Research) under Grant ex60% to the Department of Experimental and Clinical Biomedical Sciences "Mario Serio", University of Florence.

**Conflicts of Interest:** The authors declare no conflict of interest.

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
