# Peer review of "Combined Use of a Transmission Detector and an EPID-Based In Vivo Dose Monitoring System in External Beam Whole Breast Irradiation: A Study with an Anthropomorphic Female Phantom"

_applsci, doi:10.3390/app10217611_

Round 1
Reviewer 1 Report
The authors present the combined usage of the transmission detector and EPID-based in vivo dose monitoring system in whole breast irradiation. This work reports the simultaneous detection of deliver and patient errors on real time, overcoming the drawbacks of conventional pre-treatment verification method. Sensitivities of IQM signal and SoftDiso R-value are also discussed.
There are only some minor questions for the authors in order to better understand this article.
- In the abstract, authors are suggested to give a full name before using the abbreviated form “IQM”.
- What’s the measurement error range of the IQM detector itself and how is it significant when compared with the setup errors measured in this article?
- In Figure 1, authors are suggested to add annotations to show the necessary information of the measurement object and device. The right lines in the figure are not clearly shown and explained.
- Authors are suggested to give one simple schematic of optical path of the detection signal for the monitoring system in order to understand its working mechanism.
- In Figure 2, the annotation words are not clear. High resolution images are needed.
- Font size is suggested to be larger for the legends in Figures 3 – 5.
- According to the measurement results in Figure 5, phantom displacement shows obvious influences on the global gamma passing rate (γPR) and gamma mean value (γmean) indexes. If the phantom displacement always happens, is there any way to control or reduce the error rather than setting the warning threshold without further adjustment?
- What are possible differences in errors for measurements on real human body compared with the simulated results in this article?
Author Response
POINT 1 In the abstract, authors are suggested to give a full name before using the abbreviated form “IQM”.
Thank you for your comment. Full name added in Line 16 (bold). “We evaluate the combined usage of two systems, the Integral Quality Monitor (IQM)16transmission detector and the SoftDiso software, for in vivo dose monitoring by simultaneous detection of delivery and patient setup errors in whole breast irradiation.”
POINT 2 What’s the measurement error range of the IQM detector itself and how is it significant when compared with the setup errors measured in this article?
Thank you for your comment. The IQM detector error was estimated 0.08%, negligible in comparison to deviation of IQM signal consequent to small errors in delivery. A clarification added in Line 162-163 (bold). “Nevertheless IQM measurement errors are negligible compared to deviation of signal consequent to delivery errors in clinical routine.”
POINT 3 In Figure 1, authors are suggested to add annotations to show the necessary information of the measurement object and device. The right lines in the figure are not clearly shown and explained.
Authors are suggested to give one simple schematic of optical path of the detection signal for the monitoring system in order to understand its working mechanism.
As required, a schematic path of setup and annotations have been added in Figure 1 to clarify the integration of both systems.
POINT 4 In Figure 2, the annotation words are not clear. High resolution images are needed.
As required, Figure 2 (now 3) has been modified.
POINT 5 Font size is suggested to be larger for the legends in Figures 3 – 5.
As required, the legends of Figure 3-5 (now 4-6) has been modified.
POINT 6 According to the measurement results in Figure 5, phantom displacement shows obvious influences on the global gamma passing rate (γPR) and gamma mean value (γmean) indexes. If the phantom displacement always happens, is there any way to control or reduce the error rather than setting the warning threshold without further adjustment?
Thank you for your question. Imaging systems, as Cone Beam CT or Planar Portal Imaging are used to check the position of patient before the first session of treatment (and repeated for example weekly, based on specific protocols). Nevertheless these systems do not intercept deviations in patient’s position during the delivery. Thresholds in Softdiso software and IQM system are useful to detect “on treatment” errors. Specific thresholds can be set for different districts without further adjustment.
POINT 7 What are possible differences in errors for measurements on real human body compared with the simulated results in this article?
Thank you for your question. The errors simulated in this study are delivery errors and errors in patient’s setup. All kinds of errors occur in clinical routine. The scenarios can be various: small deviations can occurred, but even large errors can occurred for example when patient is not correctly identified or prepared for the therapy. The strength of the combined use of the two systems is their capability in detecting all different kinds of errors even when very small.
Sentences to clarify this issue have been added in Line 186-191 (bold).
“In this study deviations in treatment delivery and in patient’s setup were simulated. Both types of errors happen actually in clinical routine. Scenarios can be various: small, but even large deviations can occurred during the treatment, for example when a wrong linac dose calibration is done or when patient is not correctly identified or prepared for the therapy. The strength of the combined use of the two systems in detecting all different kinds of errors, even when very small, has been evidenced in this work.”

Reviewer 2 Report
The topic of the paper is the investigation of the simultaneous use of the IQM detector and SoftDiso for quality assurance in radiotherapy. This was performed on one example of 3D confromal radiotherapy of breast irradiation, testing the sensitivity of both QA detector und software to the positional errors as well as MUs and jaws-position errors.
The question is, is it possible in the radiotherapy routine that such errors happen? MUs are normaly double measured on Linac, jaws error of several centemeters are not common.
This paper needs an additional paragraph on real implementation of such QA procedures in the radiotherapy routine - when and where it makes sense to implement this procedure in clinical practice, and which kind of realistic error shold be monitored?
Author Response
response to Reviewer 2 comments:
POINT 1: The topic of the paper is the investigation of the simultaneous use of the IQM detector and SoftDiso for quality assurance in radiotherapy. This was performed on one example of 3D confromal radiotherapy of breast irradiation, testing the sensitivity of both QA detector und software to the positional errors as well as MUs and jaws-position errors.
The question is, is it possible in the radiotherapy routine that such errors happen? MUs are normaly double measured on Linac, jaws error of several centemeters are not common.
Thank you for your question. The errors simulated in this study are delivery errors and errors in patient’s setup. All kinds of errors occur in clinical routine. MUs errors mimic a wrong calibration of linac’s monitor chambers, errors in jaw’s position or multileaf collimator position can be occurred for a wrong calibration of jaw’s or collimator’s control system. 1mm-7mm deviations in jaw’s position have been analyzed. A sentence to clarify this issue has been added in Line 132-133 (bold).
“To simulate output errors,derived for example to an incorrect dose calibration of linac, MUs were modified by adding 2 MU, 3 MU, 5 MU, 10 MU……. “
POINT 2: This paper needs an additional paragraph on real implementation of such QA procedures in the radiotherapy routine - when and where it makes sense to implement this procedure in clinical practice, and which kind of realistic error shold be monitored?
Thank you for your comment. This is a pilot study simulating errors can be occurred in delivering a radiotherapy conformal breast treatment. The clinical implementation is a next step of our project.
The strength of this study is the capability of the two systems to detect all different kinds of errors even when very small. The combined use of the two detector is advantageous for example for IMRT (Intensity Modulated Radio Therapy) or VMAT (Volumetric Modulated Arc Therapy) treatments. In these cases also small deviations in positions of leaves of multileaf collimator could toughly affect the output delivery.
Two sentences have been added in Line 228-233 (bold).
“The work presented is a pilot study simulating errors which could occurr in delivering a radiotherapy conformal breast treatment. The clinical implementation and the extension of this quality protocol to other districts is the next step of our project. The combined use of the two detectors is advantageous for IMRT or VMAT treatments too. In these cases, also small deviations in positions of the multileaf collimator leaves could toughly affect the output delivery.”
